# Influence of Linewidth Enhancement Factor on the Nonlinear Dynamics and TDS Concealment of Semiconductor Ring Lasers

Yichen Wang [1,†], Xianglong Wang [1,†], Penghua Mu [1,*], Gang Guo [2], Xintian Liu [2], Kun Wang [1], Pengfei He [1], Guoying Hu [1] and Gang Jin [1]

1    School of Physics and Electronic Information, Yantai University, Yantai 264005, China;
lwb19981225@s.ytu.edu.cn (Y.W.); ytuwxl201957506211@s.ytu.edu.cn (X.W.);
wangkun2021@s.ytu.edu.cn (K.W.); hpf_972@ytu.edu.cn (P.H.); ytbyu123@s.ytu.edu.cn (G.H.);
jingang107@126.com (G.J.)
2    FISEC Infomation Technology Company Limited, Weihai 264200, China; gg87@fisherman-it.com (G.G.);
liuxintian@fisherman-it.com (X.L.)
*    Correspondence: ph_mu@ytu.edu.cn
†    These authors contributed equally to this work.

**Abstract:** In this paper, the influences of linewidth enhancement factor on the output characteristics of a semiconductor ring laser (SRL) are numerically investigated. By constructing a master–slave injection model, we discuss the influence of linewidth enhancement factor on the output characteristics of SRL. In addition, the 0–1 chaos test is introduced to study the effects of linewidth enhancement factor, feedback strength, feedback time delay and normalized injection current on the dynamic characteristics of the master laser. Furthermore, a simulation study is carried out on the suppression of time delay characteristics by the linewidth enhancement factor. The results show that selecting a proper linewidth enhancement factor has a significant effect on the chaotic output of SRL, and a larger linewidth enhancement factor is beneficial for the concealment of time delay signature. Such results are beneficial for achieving the security chaos communication and physical random generators.

**Keywords:** semi-conductor ring laser; chaos; linewidth enhancement factor; time delay signatures

## 1. Introduction

Optical chaos generated by semiconductor lasers (SLs) has attracted much attention because of its potential applications of chaos-based communications [1–4], ultrafast random generation (RNG) [5–7], compressive sensing [8], neuro-inspired signal processing [9] and chaotic radar/laser radar (LIDAR) [10,11]. Studies have shown that SLs could exhibit rich dynamics under external perturbations [12–17], including, but not limited to, conventional optical feedback, frequency selecting feedback, filtered optical feedback, opto-electronic feedback and current modulation.

Conventional optical feedback is the most commonly used method for generating chaos due to the simplest structure, but the external feedback structure introduces the time delay signature (TDS) in the chaos signals, which can be extracted by mathematical methods [18,19], such as the autocorrelation function (ACF), permutation entropy (PE) and delayed mutual information (DMI). It has been proved that the TDS could threaten the security of chaos communication and reduce the randomness of the random number generator [6,20]; thus, the suppression/concealment of TDS in the chaotic system has been the hot topic in recent years. Researchers have proved that the TDS could be removed from different chaotic systems [21–26], for example, the master–slave injection system, mutual injection system, cascade injection system and a single laser subject to different feedback configurations. Among these schemes, a master–slave injection system could achieve both TDS concealment and bandwidth enhancement.

At the very beginning, researchers only considered the influence of system parameters, including frequency detuning, coupled strength and bias current, on the concealment of TDS in the above systems. In 2015, Li found that the internal parameter of the laser, such as linewidth enhancement factor $\alpha$, has a significant effect on TDS suppression in an external feedback chaotic system and optical injection chaotic system [21]. In our previous work, the effects of $\alpha$ and non-linearity gain on TDS concealment in a mutually coupled chaotic system were studied [26]. However, the above reports are aimed at the distributed feedback semiconductor laser, while studies on the influence of internal parameters on the semiconductor ring laser (SRL) are still lacking. The SRL could output lights in two counter-propagating directions, referring to the clockwise (CW) and counter-clockwise (CCW) mode. The SRL could exhibit rich dynamics due to the variety feedback and injection path; thus, it is widely used in chaos-based communication, random number generators and optical memory.

In this paper, we study the effects of linewidth enhancement factor on the dynamics and TDS concealment of the SRL in the master–slave system. The master SRL (MSRL), subject to the self-feedback configuration, and the slave SRL (SSRL) receives the chaotic signals from the MSRL. The 0–1 chaos test is introduced to quantify the chaotic dynamics of MSRL, and ACF is hired to digitalize the TDS of MSRL and SSRL.

The paper is organized as follows. In Part 2, we discuss the theoretical model. Part 3 presents the numerical results in detail, and we finally draw our basic conclusions in Part 4.

## 2. Theoretical Model

The system structure studied in this paper is a typical master–slave injection system. The master semiconductor ring laser (MSRL) is connected with a conventional external optical feedback cavity, and it has two feedback modes, i.e., self-feedback mode and cross-feedback mode. The rate equations of the system are described using the Lang–Kobayashi equations [27], where the feedback term and injection term are introduced here. In this article, we adopt the self-feedback model, that is, *CW* feedback back to *CW* and *CCW* feedback back to *CCW*. There are also two injection methods, (1) *CW* injection to *CW*, *CCW* injection to *CCW*; (2) *CW* injection to *CCW*, *CCW* injection to *CW*. In this article, we choose the first case. These equations are shown as [28]

$$\frac{dE_M^{cw/ccw}}{dt} = \kappa(1+i\alpha)[gN_M-1]E_M^{cw/ccw} - k(1 \mp \delta_k)e^{i\phi}E_M^{ccw/cw} + \eta E_M^{cw/ccw}(t-\tau_f)e^{i\theta} \quad (1)$$

$$\frac{dE_S^{cw/ccw}}{dt} = \kappa(1+i\alpha)[gN_S-1]E_S^{cw/ccw} - k(1 \mp \delta_k)e^{i\phi}E_S^{ccw/cw} \\ + k_{inj}E_M^{cw/ccw}(t-\tau_{inj})\exp(-i\omega_M\tau_{inj}-i2\pi\Delta ft) \quad (2)$$

$$\frac{dN_{M/S}^{cw/ccw}}{dt} = \gamma_N[\mu - N_{M/S}^{cw/ccw} - g_{M/S}^{cw/ccw}N_{M/S}^{cw/ccw}|E^{cw/ccw}|^2 - g^{ccw/cw}N_{M/S}^{cw/ccw}|E^{ccw/cw}|^2] \quad (3)$$

$$g_{M/S}^{cw/ccw} = 1 - s|E_{M/S}^{cw/ccw}|^2 - c|E_{M/S}^{cw/cw}|^2 \quad (4)$$

where *M* and *S* stand for MSRL and SSRL, respectively, and *E*(t) is the electric field. *N*(t) denotes the carrier density. The last term of Equation (1) is the optical feedback of MSRL; the last term of Equation (2) is the injection term from MSRL to SSRL. $\kappa = 100 \text{ ns}^{-1}$ is the field decay rate, $\alpha$ is the linewidth enhancement factor, *g* is the differential modal gain, $k = 0.44 \text{ ns}^{-1}$ is the back scattering rate, $\delta_k = 0.2$ is the asymmetry factor, $\phi = 1.5$ is the phase shift, $\omega_M$ is the angular frequency of MSRL. $\gamma_N = 0.2 \text{ ns}^{-1}$ is the carrier decay rate, $\mu$ is the re-normalized injection current, $s = 0.005$ is the self-saturation coefficient, $c = 0.01$ is the cross-saturation coefficient. The feedback parameters contain $\eta$, $\tau_f$ and $\theta$, which denote feedback strength, feedback delay and constant feedback phase, respectively. The injection parameters include $k_{inj}$, $\tau_{inj}$ and $\Delta f$, which stand for injection strength, injection delay and frequency detuning between MSRL and SSRL.

As mentioned above, the TDS could be extracted through ACF, which is defined as [18]

$$C(s) = \frac{< [x(t)- < x(t) >][x_s(t)- < x(t) >] >}{< [x(t)- < x(t) >]^2 >} \tag{5}$$

where $x(t)$ represents time series, $s$ is the shift, $< \cdot >$ stands for the average, and the delayed form of the original time series is $x_s(t) = x(t - s)$.

## 3. Numerical Results

In this work, we firstly focus on the influence of $\alpha$ on the dynamics of SRL; then, the TDS concealment in the master–slave injection system is discussed. The rate equations mentioned above were simulated using the famous fourth-order Runge–Kutta algorithm (RK-4).

### 3.1. Dynamical Mappings of MSRL

Figure 1 shows the dynamics of MSRL with the varying parameters, including external parameters, such as feedback strength $\eta$, bias current $\mu$, feedback delay $\tau_f$ and internal parameter, i.e., linewidth enhancement factor $\alpha$. We hire the 0–1 chaos test to quantify the dynamics of MSRL, while the chaotic region is highlighted in dark red, and the dark blue represents the steady state. In Figure 1, the X-axis denotes $\mu$, and the Y-axis stands for $\eta$; Figure 1(a1–a4) are the CW outputs of MSRL, Figure 1(b1–b4) are the CCW outputs of MSRL, Figure 1(a1,a2,b1,b2) show that the $\tau_f = $ 6ns, and Figure 1(a3,a4,b3,b4) show that the $\tau_f = $ 3ns. In addition, Figure 1(a1,b1,a3,b3) indicate $\alpha = $ 2.5, and Figure 1(a2,b2,a4,b4) stand for $\alpha = $ 5. It can be found from Figure 1 that with the increase in $\mu$, bigger $\eta$ is required to ensure that MSRL works at the chaotic oscillation. When we choose the smaller linewidth enhancement factor, i.e., $\alpha = $ 2.5, by comparing Figure 1(a3,b3) $\tau_f = $ 3ns and Figure 1(a1,b1) $\tau_f = $ 6ns, we can find that strong feedback strength is helpful in obtaining the chaos. When the linewidth enhancement factor has a large value, such as $\alpha = $ 5, the chaotic region remains almost unchanged in a different feedback delay, i.e., Figure 1(a2,b2) $\tau_f = $ 3ns and Figure 1(a4,b4) $\tau_f = $ 6ns. That is to say, with the increase in $\alpha$, the effect of feedback delay on the output characteristics of MSRL is weakened. Meanwhile, by comparing two different $\alpha$, such as Figure 1(a1,a2), it can be found that the bigger the linewidth enhancement factor $\alpha$, the larger the region of chaos that can be obtained in MSRL.

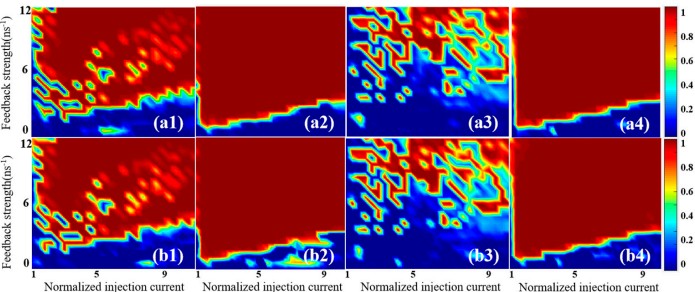

**Figure 1.** The two-dimensional map of the 0–1 test for chaos of MSRL in the ($\mu$, $\eta$) plane. (a1,a2,a3,a4): 0–1 test of the CW outputs, (b1,b2,b3,b4): 0–1 test of the CCW outputs; (a1,b1,a2,b2): $\tau_f = $ 6 ns, (a3,b3,a4,b4): $\tau_f = $ 3 ns; (a1,b1,a3,b3): $\alpha = $ 2.5, (a2,b2,a4,b4): $\alpha = $ 5.

Figure 2 shows the 0–1 chaos test of MSRL, while the X-axis and the Y-axis are the injection current and feedback delay time, respectively. Figure 2(a1,b1,a2,b2) represent the strong feedback $\eta = $ 4 ns$^{-1}$, and Figure 2(a3,b3,a4,b4) represent the weaker feedback $\eta = $ 2.5 ns$^{-1}$. We set a small value of linewidth enhancement factor $\alpha = $ 2.5 in Figure 2(a1,b1,a3,b3) and a bigger linewidth enhancement factor $\alpha = $ 5 in Figure 2 (a2,b2,a4,b4). In the cases of weakened feedback strength, the dark red focuses on the left region of the two-dimensional maps, which means that with too big a value of the injection current, the MSRL is not operating in the chaotic region. From Figure 2(a2,b2), we can find that by choosing bigger $\eta$ and $\alpha$, the MSRL outputs chaos in the whole parameter space

of $(\mu, \tau_f)$, that is to say, the effects of the injection current and feedback delay time on the dynamics of MSRL can be weakened by increasing $\eta$ and $\alpha$.

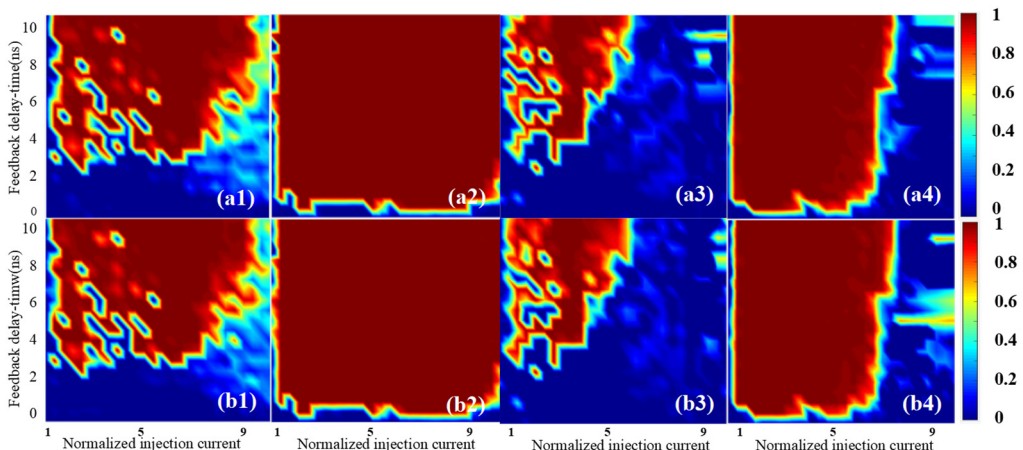

**Figure 2.** The two-dimensional map of the 0–1 test for chaos of MSRL in the $(\mu, \tau_f)$ plane. (a1,a2,a3,a4): 0–1 test of the CW outputs, (b1,b2,b3,b4): 0–1 test of the CCW outputs; (a1,b1,a2,b2): $\eta = 4\,\mathrm{ns}^{-1}$, (a3,b3,a4,b4): $\eta = 2.5\,\mathrm{ns}^{-1}$; (a1,b1,a3,b3): $\alpha = 2.5$, (a2,b2,a3,b4): $\alpha = 5$.

### 3.2. TDS Suppression

Until now, we studied the influence of linewidth enhancement factor on the dynamics of MSRL. In this subsection, we will focus on TDS concealment in the master–slave injection system.

Figure 3 shows the timeseries and the corresponding spectrum and ACF of MSRL for several but different $\alpha$. The feedback parameters are set as the feedback strength $\eta = 4\,\mathrm{ns}^{-1}$, feedback delay time $\tau_f = 6\,\mathrm{ns}$, and current $\mu = 2.5$. Under the above conditions, we can find that the MSRL outputs the chaotic fluctuations. In the right row, the obvious peak around the delay time of MSRL, which is extracted by ACF, can be observed. The ACF peak is seen very clearly when $\alpha = 3.5$; as $\alpha$ increases to 5, the ACF peak is slightly attenuated. Generally speaking, the ACF can be considered effectively suppressed, while the value is less than 0.2. In Figure 3(c3), the chaos TDS is strongly weakened, while $\alpha = 8$. The reason is due to the coupling effect between the amplitude and the phase of the electric fields. Interestingly, as $\alpha$ increases, the spectrum of MSRL expands. The TDS value of MSRL is still bigger than 0.2 in Figure 3(c3). Then, we propose a master–slave chaotic system to suppress the TDS in slave SRL (SSRL). Figure 4 represents the simulated results of timeseries, spectrum and ACF of SSRL, while the system parameters are set as: $\Delta f = 10\,\mathrm{GHz}$, $k_{\mathrm{inj}} = 35\,\mathrm{ns}^{-1}$. The ACF of SSRL is lower than MSRL; for example, in the case of $\alpha = 3.5$, the value of ACF is 0.67, while the ACF peak size of SSRL is 0.3. The effect of TDS suppression on the injection system has an obviously good performance, while the linewidth enhancement factor of MSRL and SSRL grows to 8, which can be seen from Figure 3(c3) and Figure 4(c3). The ACF value of MSRL is 0.24, and TDS is totally concealed in SSRL. The bandwidth of SSRL is also bigger than MSRL, which can be seen from Figure 4(a2–c2).

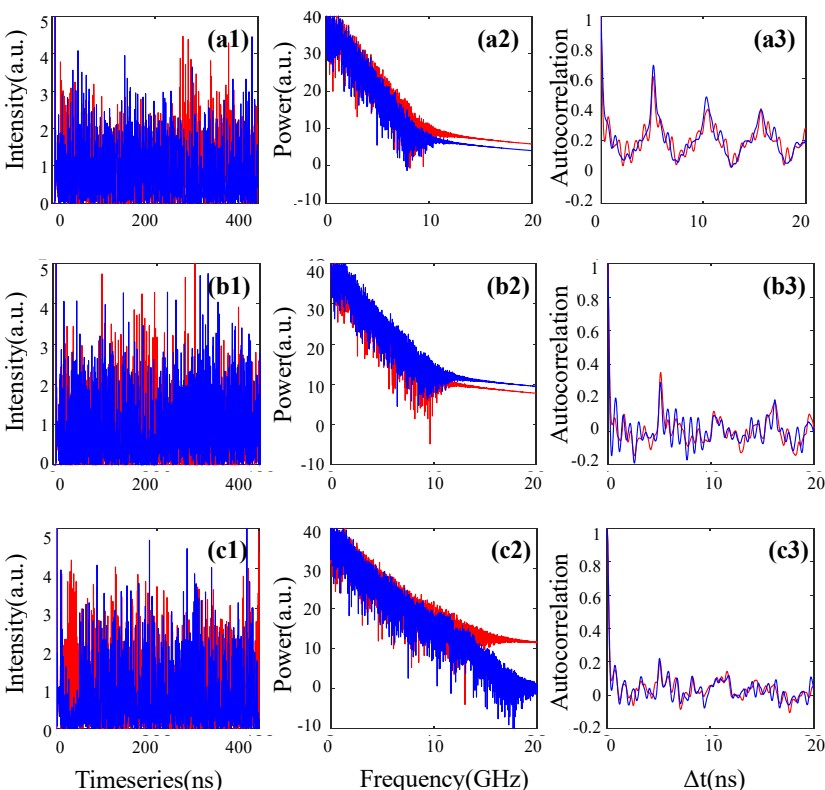

**Figure 3.** The timeseries (**a1–c1**), power spectrum (**a2–c2**) and ACF (**a3–c3**) of the MSRL, where (**a1–a3**) $\alpha = 3.5$, (**b1–b3**) $\alpha = 5$, and (**c1–c3**) $\alpha = 8$. The blue line denotes CW, and the red line denotes CCW. The parameters are set as: $\eta = 4\,\mathrm{ns}^{-1}$, $\tau_f = 6\,\mathrm{ns}$, and $\mu = 2.5$.

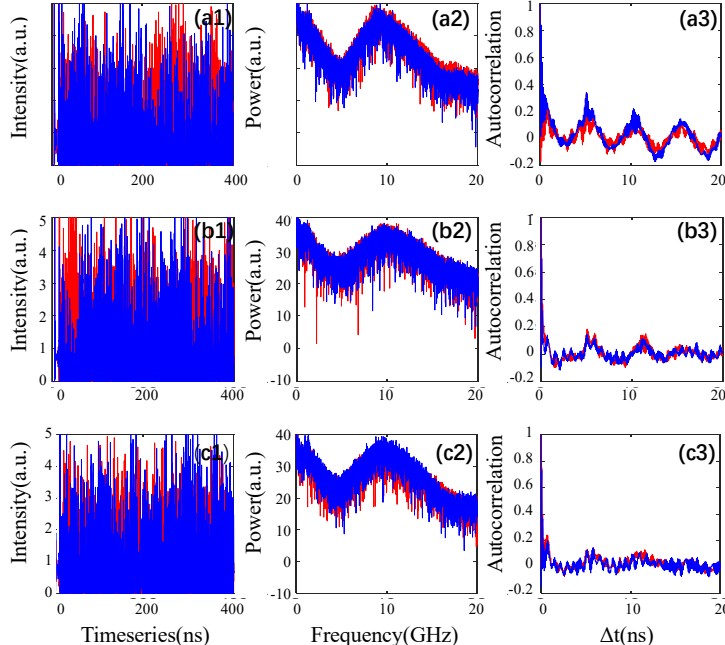

**Figure 4.** The timeseries (**a1–c1**), power spectrum (**a2–c2**) and ACF (**a3–c3**) of the SSRL. The blue line denotes CW, and the red line denotes CCW. The currents of MSRL and SSRL are $\mu = 2.5$. The parameters are set as: $\Delta f = 10\,\mathrm{GHz}$, $k_{\mathrm{inj}} = 35\,\mathrm{ns}^{-1}$. First row: $\alpha = 3.5$, second: $\alpha = 5$, third: $\alpha = 8$.

Figure 5 shows the curve of TDS peak size in the variation range of frequency detuning between MSRL and SSRL, where the injection strength is fixed at $k_{inj} = 35$ ns$^{-1}$. In Figure 5, the peak size is defined as the maximum value of ACF during the time window $[\tau_f - 0.05\tau_f, \tau_f + 0.05\tau_f]$. From Figure 5, we can find the red line runs on the top and is followed by the black line, the blue line and the green line, which means higher $\alpha$ leads to better TDS concealment in SSRL. In the case of smaller $\alpha$, i.e., the red line ($\alpha = 2.5$) and the black line ($\alpha = 3.5$) in Figure 5, the peak size becomes substantially larger with the condition of small frequency detuning, i.e., $-20$ GHz $< \Delta f < 20$ GHz. The value of the ACF peak size is less than 0.2 over the whole range of $\Delta f$, while the SRLs have large $\alpha$, i.e., the green line ($\alpha = 8$) in Figure 5, which means the large $\alpha$ widens the choice of $\Delta f$ for the purpose of hiding TDS.

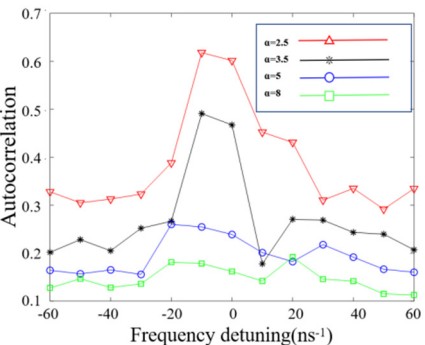

**Figure 5.** The line diagram of TDS as the frequency detuning between MSRL and SSRL varying from $-60$ GHz to 60 GHz, while red triangles denote $\alpha = 2.5$, black stars denote $\alpha = 3.5$, blue circles denote $\alpha = 5$, green squares denote $\alpha = 8$.

Next, we study further the influence of $\alpha$ on TDS concealment during the range of $k_{inj}$ in Figure 6, while the $\Delta f$ is fixed at $\Delta f = 5$ GHz. In Figure 6, we can find that the trends for ACF peak size of SSRL in different $\alpha$ are the same as the results of Figure 5, and larger $\alpha$ is favorable for suppressing the time delay signature of SSRL over a wide range of injection strength. It is interesting to find that the MSRL and SSRL can achieve chaos synchronization when the injection strength is large enough.

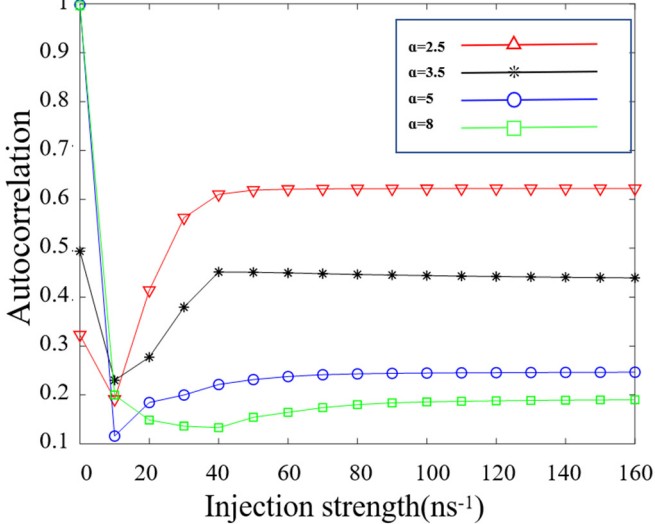

**Figure 6.** The line diagram of TDS as the injection strength varying from 0 to 160 ns$^{-1}$, where red triangles denote $\alpha = 2.5$, black stars denote $\alpha = 3.5$, blue circles denote $\alpha = 5$, green squares denote $\alpha = 8$.

## 4. Conclusions

In this paper, we numerically investigated TDS concealment in one master–slave chaotic system constructed with two SRLs. By hiring the 0–1 chaos test and ACF, we discussed the influence of $\alpha$ on the chaos characteristics of SRLs, including the nonlinear dynamics and time delay signature. The results indicate that the larger $\alpha$ can not only ensure the MSRL generates the chaos with high complexity, but it also has a favorable effect on TDS concealment in SSRL. The simulation results also indicate that one can reduce the dependence of chaos on feedback parameters by choosing the big value of $\alpha$. The findings pave the way for chaos-based applications, such as chaotic communication, chaotic radar and random number generators.

**Author Contributions:** Methodology, Y.W., G.H. and P.M.; validation, X.W. and P.M.; investigation, Y.W., G.J. and P.H.; writing—original draft preparation, X.W. and G.G.; writing—review and editing, P.M., X.L. and K.W. All authors have read and agreed to the published version of the manuscript.

**Funding:** This work was supported by the Project: Natural Science Foundation of Shandong Provincial (ZR2020QF090), The Key Lab of Modern Optical Technologies of Education Ministry of China, Soochow University (KJS2066); The Key Lab of Advanced Optical Manufacturing Technologies of Jiangsu Province, Soochow University (KJS2045).

**Institutional Review Board Statement:** Not applicable.

**Informed Consent Statement:** Not applicable.

**Data Availability Statement:** Not applicable.

**Acknowledgments:** The authors would like to thank all reviewers for their helpful comments and suggestions on this paper.

**Conflicts of Interest:** The authors declare no conflict of interest.

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
