# Peer review of "Influence of Linewidth Enhancement Factor on the Nonlinear Dynamics and TDS Concealment of Semiconductor Ring Lasers"

_electronics, doi:10.3390/electronics11132007_

Round 1

Reviewer 1 Report

The author numerically investigated the influences of the linewidth-enhancement factor on the output characteristics of semiconductor ring laser (SRL). The manuscript is written well. However, it needs to be improved. Therefore, I recommend the authors address the following issues before it goes for publication.

1. Please provide the full form of any acronyms used in the manuscripts the very first time you use them in the manuscript. Moreover, there are a few acronyms whose full form has not been mentioned anywhere in the manuscript, such as LIDAR.

2. page 2, line 66: please correct the spelling of semiconductor.

3. Out of the two external optical cavity feedback modes to the MSRL, the authors considered only the self-feedback mode and have not talked anything about the cross-feedback. It is unclear why the authors chose the self-feedback mode only and discarded the cross-feedback.

4. Page 2, line 74: please delete one of the “Where E(t) is”. The phrase is repeated.

5. Please mention what M and S stand for on page 2, line 75.

6. Please cite proper references for equations (1) to (5).

7. page 4, line 115: please correct spelling of respectively.

8. page 4, line 128: replace “dynamics of MSRL, in this subsection,” with “dynamics of MSRL. In this subsection,”.

9. page 5, line 135, Fig.4 caption, please replace c_2 with c2.

10. page 6, line 148, please replace interesting with interestingly.

11. page 6, line 160, please replace while with where.

12. With the help of Fig.6, the authors concluded that the larger a is favorable for suppressing the time delay signature of SSRL. But it is hard to make such a bold conclusion from only four values of alpha. It would be great if the authors could add a few more curves with a few more values of alpha, one in between 5 and 8 and others greater than 8. I will be totally fine to have them in the supplemental material.

13. page 7, line 180: please correct the superscript typo in 160 ns-1.

14. page 7, line 180: please replace while with where.

15. In conclusion, the authors say, “ The results indicate that the larger a cannot only ensures the MSRL generates the chaos with high complexity, but also has a favorable effect on the TDS concealment in SSRL.” My questions to the authors are (a) How big is big for alpha? (b) What is the maximum possible value of alpha?  

16. page 7, line 187: please replace can not with cannot.

17. Is the first sentence of the acknowledgments necessary?

Author Response

Thank you for your positive comments. We have made some changes to our manuscript according to the comments of the other reviewer and hope you will continue to support our work.

Reviewer 2 Report

In this paper the effects of the alpha factor on the chaotic dynamics of a semiconductor ring laser are studied numerically. Both the single laser and the master-slave system are analyzed.

I have several doubts about this paper, here are the main ones.

Theoretical model. In the equations for the electric fields the backscattering term should couple CW with CCW and vice versa, instead now it is a re-injection term; the asymmetry factor delta_k is useless because it just rescales the factor k; in the exponential of Eq. (2) I think that the exponent should be simply 2*pi*Delta*f*tau_inj, in any case an “i” is missing in the term omega_M*tau_inj; in Eq. (2) the feedback term does not appear, why? Are not the two ring lasers equal? ; in Eq.(3) the subscripts M,S are missing in the gain factors and the field intensity should appear instead of the modulus. The parameters mu and theta are not defined

Numerical results. I think that there is an inconsistency between Figs. 1 and 2. In Fig. (a4), alpha=5 and tau_f=3 ns, and for eta=2.5 1/ns and large mu the 0-1 chaos test gives chaos; in Fig. (b4), alpha=5 and eta=2.5 1/ns, and for tau_f=3 ns and large mu the 0-1 chaos test gives non-chaotic dynamics.

A major comment is that the main effects in Figs. 4 to 6 are found for alpha=8 which is quite an unrealistic value. Moreover, I guess that the peak at 10 GHz in the power spectrum is detrimental for chaos as the one in the auto correlation function at tau_f.

For all these reasons I think that the paper is not suitable for publication.

Author Response

Thank you very much for careful reading of our manuscript. Our reply to your questions and suggestions is detailed as follows.

Round 2

Reviewer 1 Report

There are a few inconsistencies in font size and font family in the mathematical quantities presented in the manuscript. These inconsistencies are either resulted during the pdf conversion or they inherently there in the manuscript. Assuming that these inconsistencies will be fixed during proofreading and English language check, I am ok with this version of the manuscript.

Author Response

Thank you for your positive comments. We hope you will continue to support our work.

Reviewer 2 Report

In the revised version of the manuscript the authors have corrected the model equations and Fig. 1. Yet two important issues remain.

The authors claim that the parameter alpha in semiconductor lasers ranges 1 to 10, but I have never seen reported values larger than 5 or 6. Could they provide at least one experimental reference?

Could the authors explain in which sense a peak at 10 GHz in the spectrum "has no periodicity". I would say that it is associated with a period of 0.1 ns.

Author Response

Thank you for your suggestions. We hope you will continue to support our work.

Round 3

Reviewer 2 Report

The authors have provided the requested references about big values of the alpha factor and explained why in their opinion the peak at 10 GHz associated with the beating frequency is not detrimental for secure chaotic communications.

This manuscript is a resubmission of an earlier submission. The following is a list of the peer review reports and author responses from that submission.